# Yolov8n-FADS: A Study for Enhancing Miners’ Helmet Detection Accuracy in Complex Underground Environments

**DOI:** 10.3390/s24123767

**Published:** 2024-06-10

**Authors:** Zhibo Fu, Jierui Ling, Xinpeng Yuan, Hao Li, Hongjuan Li, Yuanfei Li

**Affiliations:** School of Coal Engineering, Shanxi Datong University, Datong 037000, China; 220857002126@sxdtdx.edu.cn (Z.F.); 230857002122@sxdtdx.edu.cn (J.L.); 230857002112@sxdtdx.edu.cn (H.L.); 210857002137@sxdtdx.edu.cn (H.L.); 220857002116@sxdtdx.edu.cn (Y.L.)

**Keywords:** Yolov8n, miners’ helmet recognition, detection head, attention mechanism, loss function

## Abstract

A new algorithm, Yolov8n-FADS, has been proposed with the aim of improving the accuracy of miners’ helmet detection algorithms in complex underground environments. By replacing the head part with Attentional Sequence Fusion (ASF) and introducing the P2 detection layer, the ASF-P2 structure is able to comprehensively extract the global and local feature information of the image, and the improvement in the backbone part is able to capture the spatially sparsely distributed features more efficiently, which improves the model’s ability to perceive complex patterns. The improved detection head, SEAMHead by the SEAM module, can handle occlusion more effectively. The Focal Loss module can improve the model’s ability to detect rare target categories by adjusting the weights of positive and negative samples. This study shows that compared with the original model, the improved model has 29% memory compression, a 36.7% reduction in the amount of parameters, and a 4.9% improvement in the detection accuracy, which can effectively improve the detection accuracy of underground helmet wearers, reduce the workload of underground video surveillance personnel, and improve the monitoring efficiency.

## 1. Introduction

Ensuring the personal safety of workers involved in mining operations is paramount. A primary safeguard for underground coal miners is the utilization of helmets, offering crucial head protection during emergencies, thus preserving lives. However, lapses in helmet usage by miners do transpire, posing risks to mine safety. Given the pivotal role of helmets in safeguarding workers, detecting compliance with helmet regulations emerges as a vital aspect of production management. Yet, conventional manual oversight proves inefficient. Hence, investigating helmet detection algorithms holds considerable importance for advancing mine safety intelligence. As artificial intelligence technology advances continuously, the day-to-day oversight of underground personnel has shifted toward an information- and intelligence-oriented approach. Preventing unsafe behavior among underground personnel is considered the primary method for mitigating mining accidents [1]. Currently, underground video monitoring remains predominantly reliant on manual operation [2]. However, the fatigue experienced by long-term monitors on duty is inevitable, hindering the capacity to fulfill the need for real-time and precise monitoring. Hence, the adoption of AI monitoring techniques holds promise in minimizing manual intervention, alleviating the workload of personnel, enhancing the monitoring efficacy, and consequently, diminishing the incidence of mining accidents.

In the contemporary epoch of deep learning, computer vision technology has emerged as a prominent research domain, particularly in image processing analysis [3,4]. Research aimed at identifying unsafe behavior among subterranean workers primarily concentrates on two principal facets: ensuring the proper utilization of safety equipment and monitoring workers’ conduct. The accurate usage of safety gear is paramount for subterranean workers, prompting researchers to leverage convolutional neural networks and other methodologies to augment the speed and precision of helmet usage detection. Qirui Li [5] initially delineated the human body region through directional gradient histograms, extracted head information utilizing ‘convex’ head features, and amalgamated gradient histogram (HOG) and support vector machine (SVM) techniques for helmet detection. Sun et al. [6] employed a visual background difference algorithm to identify workers, determined the helmet position based on the head-to-body proportions, and applied a Bayesian optimization-based SVM model for helmet detection. Concurrently, Li Tian et al. [7] utilized the Visual Background Extraction (ViBe) algorithm for background modeling, segmented the images based on moving targets, employed a real-time human classification framework for pedestrian localization, and utilized the head position, color space transformation, and color features for helmet usage detection.

Deep learning-based methods for target detection are commonly categorized into two primary types based on the detection stage: two-stage detection and single-stage detection [8,9]. Two-stage detection algorithms bifurcate the detection process into two distinct stages: initially, a candidate image generation network conducts feature extraction, which is then followed by the utilization of a classification network for categorization and refinement of the edge positions. Among the two-stage detection approaches, Girshick et al. [10] proposed a technique based on the region convolutional neural network R-CNN, leveraging the CNN for feature extraction and classification, thereby accomplishing target detection through region analysis. To address the challenges associated with the fixed-size fully connected layers in an R-CNN, leading to information distortion and moderate training speeds, subsequent advancements were introduced, including SPP-Net [11], FastR-CNN [12], and FasterR-CNN [13]. Notably, FasterR-CNN stands out for its performance enhancements: the incorporation of a pyramidal pooling layer post-convolutional layer enables the mapping of convolutional features of varying sizes to fixed-size fully connected inputs. Simultaneously, a region proposal network (RPN) is introduced to supplant the original selective search method (SS) [14], enhancing the speed of the candidate box generation. This two-stage detection methodology effectively segregates tasks and achieves exceptional detection accuracy.

Single-stage detection techniques, exemplified by the Yolo series, simplify object detection into a regression task, wherein an image is inputted into a neural network, directly yielding an output encompassing both category and location details. Tianyu Li et al. [15] enhanced helmet detection in intricate environments through the extension of Yolov4. Jianbo Wang [16] optimized Yolov4-tiny, tailoring it to be more suitable for edge devices by reducing its weight. Xiyu Li et al. [17] addressed the challenge of the low accuracy and the propensity to overlook personnel and helmet detection in the intricate settings of underground integrated mining faces by refining Yolov5s. Yang Yongbo et al. [18] introduced an enhanced lightweight helmet detection algorithm, Yolo-M3, which mitigates the issue of the low discrimination of occluded targets, thereby improving detection. Li Shengli et al. [19] refined Yolov7-tiny, termed DCS-Yolo, to ameliorate the leakage and false detection of miniature and occluded targets. Sun Chi et al. [20] achieved a detection speed of 28.04 frames/s on the Jetson TX2 edge device by enhancing Yolov7-tiny to fulfill real-time demands.

The previously mentioned methodologies have indeed improved the speed and accuracy of helmet detection. However, they overlook the challenging conditions prevalent on mine floors, such as the low lighting, cluttered environments, uncertain surveillance, variable surveillance distances, and the diverse appearances of helmets. Consequently, there is still room for improvement in target detection. To tackle these challenges, we have devised an enhanced algorithm for detecting helmet wear in underground environments. Dataset enhancement involves simulating occlusion, varying the lighting conditions, and introducing random blurring. Building upon Yolov8n, enhancements were made to the backbone, head, and detection mechanisms, with fine-tuning of the loss function. Additionally, an attention mechanism was incorporated. These enhancements extend beyond helmet detection and can be broadly applied to other target detection tasks.

## 2. Yolov8 Algorithm

The Yolo algorithm partitions the image into multiple grids, predicts the bounding boxes within each grid along with the classes of objects they contain, and resolves any overlapping bounding boxes using a non-maximum suppression (NMS) algorithm. As a quintessential single-stage detection algorithm, Yolo has evolved into Yolov8, the latest iteration in the Yolo lineage, introduced in 2023 by Ultralytics [21]. Noteworthy among its features is its robust scalability. Serving as a framework, Yolov8 facilitates seamless compatibility with preceding Yolo versions, enabling straightforward comparison of their performance [22]. The network structure of Yolov8n is depicted in Figure 1.

## 3. Yolov8n-FADS Detection Model

Enhancing the Yolov8n model by integrating the Dilated Reparam Block from UniRepLKNet and subsequently introducing RepNCSPELAN into Yolov9 to replace the C2f module for the backbone network leads to improvements in the convergence rate and a reduction in the parameters and computations without sacrificing accuracy. Modifications to the HEAD component involve employing attention-scale sequence fusion and introducing a P2 detection layer to refine the network structure. SEAMHead incorporates occlusion-aware Attention to enhance performance and adeptly handle occlusion scenarios. The Focaler IoU series replaces the original loss function to expedite convergence and yield more precise regression results. Additionally, Triplet Attention is integrated to enhance the feature fusion capacity of the network. The refined network structure is depicted in Figure 2.

### 3.1. Backbone Structure

The Backbone of Yolov8 is enhanced by integrating the Dilated Reparam Block from UniRepLKNet [23], a technique subsequently adopted in RepNCSPELAN within Yolov9 for further innovation. During the model design, expansion convolution is utilized to capture sparse features. Sijie Zhao proposed the DRB module (depicted in Figure 3), which incorporates parallel inflation convolution alongside large kernel convolution. Leveraging structural re-parameterization, the entire block can be equivalently transformed into a large kernel convolution. This arises from the combination of a small kernel with inflated convolution being equivalent to a large kernel with non-inflated convolution. Such a design enables the model to efficiently capture sparsely distributed spatial features, thereby enhancing its capability to discern intricate patterns. RepNCSPELAN [24] serves as a feature extraction module in Yolov9, akin to the C2f and C3 modules in Yolov5 and v8.

### 3.2. Head Structure

The enhancement of the Head component is attained through the implementation of Attentional Scale Sequence Fusion (ASF) in ASF-YOLO, along with the integration of the P2 detection layer and optimization of the network architecture. ASF-YOLO (depicted in Figure 4) represents a novel YOLO framework introduced by Ming Kang [25], amalgamating spatial and scale features to achieve precise and rapid cell instance segmentation. This framework extends the Yolo segmentation model, incorporating the Scale Sequence Feature Fusion (SSFF) module to bolster the network’s multi-scale information extraction capacity and the Triple Feature Encoder (TFE) module to fuse feature maps across various scales for detailed information augmentation. Moreover, a Channel and Position Attention Mechanism (CPAM) is introduced to synergize the SSFF and TFE modules, emphasizing information channels and spatial position-dependent small objects. The SSFF and TFE modules contribute to the enhanced multi-scale and small object instance segmentation performance, while the channel and position attention mechanism further exploits the feature information from these modules.

### 3.3. Loss Functions

The issue of the training sample imbalance persists in boundary box regression. Traditional approaches to mitigating this imbalance involve sampling and reweighting challenging samples during training, albeit with limited effectiveness. Focal Loss [26] addresses this by emphasizing the easily identifiable negative samples in the total loss and adjusting the slope accordingly. This approach enhances the model’s capacity to discern rare target categories by modulating the weights of positive and negative samples. It prioritizes difficult-to-classify positive samples while diminishing the weights of relatively easy-to-classify negative samples.

The target detection complexity is categorized into difficult and easy samples. Analyzing the target size, detectable targets are simple samples, while very small targets pose localization challenges, classifying them as difficult samples. In tasks primarily featuring simple samples, concentrating on them during bounding box regression enhances detection. Conversely, tasks abundant in difficult samples necessitate focusing on them during bounding box regression to enhance performance.

To address varied detection tasks across different regression samples, we employ a linear interval mapping method to refine the Intersection over Union (IoU) loss, thereby enhancing the boundary box regression. The formula is as follows:$$IoU^{focaler}=\left\{\begin{array}{ll} 0, & IoU<d\\ \frac{IoU-d}{u-d}, & d\ll IoU\ll u\\ 1, & IoU>u \end{array}\right\}$$
where *IoU^focaler^* is the reconstructed Focaler IoU, IoU is the original IoU value, and *d*,*u* take on the values [0, 1]. By adjusting the values of *d* and *u*, the *IoU^focaler^* can be made to focus on different regression samples. Its losses are defined as follows:$$L_{Focaler-IoU}=1-IoU^{focaler}$$

### 3.4. Detection Head

Occlusion often results in misalignment, local aliasing, and the absence of key features. To address this challenge, Ziping Yu [27] introduced the SEAM attention module and devised exclusion loss as a solution. Incorporating a multi-headed attention network, termed the SEAM module (Figure 5), facilitates multi-scale detection by accentuating image regions, thereby minimizing the background interference. Depth-separated convolution operates on a depth-by-depth principle, akin to channel-separated convolution. While this approach acknowledges the significance of different channels and reduces the parameter count, it overlooks meaningful inter-channel relationships. To rectify this limitation, the outputs from various depth convolutions are amalgamated through point-wise (1 × 1) convolution. Subsequently, a two-layer fully connected network integrates the channel information, enhancing the inter-channel connections. The knowledge acquired regarding occluded and unobstructed surfaces in the preceding step can offset the losses during occlusion. The output logarithm of the connectivity layer undergoes processing to expand the value range from [0, 1] to [1, e]. This exponential normalization establishes a monotonic mapping, enhancing the tolerance to positional errors. Ultimately, the SEAM module’s output serves as attention, augmenting the original features, thereby enhancing the head detection and bolstering the occlusion problem’s resolution capabilities.

### 3.5. Attention Mechanisms

Diganta Misra [28] introduced a nearly parameter-free attention mechanism to delineate channel and spatial attention, which is termed Triplet Attention (Figure 6). Triplet Attention comprises three parallel branches, two of which capture cross-dimensional interactions involving channel C and space H or W. The final branch, akin to CBAM, constructs Spatial Attention. These outputs from the last three branches are aggregated through averaging.

The model comprises three branches: the first computes the channel attention by passing the input features through Z-Pooling, 7 × 7 convolution, and a sigmoid activation function to generate the spatial attention weights. In the second branch, channel C and the spatial W dimensions interactively capture the features. Here, the input features undergo permutation and transformation into H × C × W dimensions, followed by H-dimensional Z-pooling and subsequent operations. This results in C × H × W dimensional features, facilitating element-wise summation. The third branch involves the C channel interacting with the H-dimensional capture branch. Initially, the input features undergo permutation to obtain W × H × C dimensional features, followed by W-dimensional Z-pooling and similar subsequent processing. The output features of all three branches are then summed to derive the average.

The Z-pool layer reduces the C-dimensional tensor to two dimensions and links the average pooled feature in that dimension to the maximum pooled feature. It can be expressed by the following equation:$$Z-pool(\chi )=[MaxPool_{0d}(\chi ),AvgPool_{0d}(\chi )]$$
where 0d is the 0th dimension in which the maximum pooling operation takes place and the average pooling operation takes place.

## 4. Experiments and Results

### 4.1. Datasets

The dataset was taken from the public dataset CUMT-HelmeT available from the China University of Mining and Technology [29,30]. The dataset was filtered and the images were labelled using Labellmg software. The training set, test set, and validation set were divided from 900 images in the ratio of 8:1:1. In this study, the Random Brightness Contrast, Gaussian Noise, and Grid Distortion from the online data enhancement library albumentations were used to improve the robustness and generalization of the dataset, taking into account the different lighting, occlusion, image quality and other different disturbances.

### 4.2. Experimental Equipment and Evaluation Indicators

The model was primarily developed in Python, utilizing the PyTorch deep-learning framework, and accelerated training was facilitated through CUDA11.3. The hardware configuration for testing comprised a 13th Gen Intel(R) Core (TM) i5-13400F CPU and an NVIDIA RTX3060Ti GPU equipped with 8 GB of video memory. During training, the input image size was set to 640 × 640, employing Stochastic Gradient Descent (SGD) as the optimization function. The training regimen consisted of 300 epochs, with a batch size of 16 and an initial learning rate of 0.01.

The model evaluation metrics used mAP as the final evaluation metric, which measureed the correct recognition rate of the model. GFLOPs were used to measure the complexity of the model or algorithm, while Params indicated the size of the model. Typically, the smaller the Params and GFLOPs, the less computation required by the model. The complexity of the model was measured by the number of model parameters (Params) and the amount of computation (GFLOPs).

### 4.3. Results of the Experiment

#### 4.3.1. Analysis of the Effectiveness of Improved Attention Mechanisms

To assess the efficacy of the fused Triplet Attention mechanism module, we employed Yolov8n as the foundational network. The MLCA, SE, SimAM, and Triplet Attention mechanisms were integrated between the C2f and SPPF modules, positioned at the terminus of the backbone network for training and validation purposes. For a comprehensive comparison of objective metrics, refer to Table 1.

#### 4.3.2. Analysis of Experimental Results

To validate the detection performance of the enhanced algorithm, conducting a detection comparison among the refined mini-modules offered a clearer insight into the contribution of each module to the improved algorithm. The test results of the model are shown in Table 2.

The experimental results in Table 2 illustrate varying degrees of improvement in the mean Average Precision (mAP) achieved by four enhancements to the Yolov8n algorithm during training compared to the original Yolov8 algorithm. The Yolov8n-F algorithm, substituting the CIoU loss function with the Focaler-IoU loss function, enhances the mAP by 1.7% while maintaining an identical model size, parameter count, and computational load, suggesting a higher detection accuracy with Focaler-IoU. Utilizing the Yolov8n-A-P2 algorithm for the head part refinement reduces the parameter volume by 0.51 M and the model size by 0.8 MB, resulting in a 4.9% increase in the mAP. This implies that the use of the Yolov8n-A-P2 algorithm can enhance the detection accuracy of the model while simplifying its complexity. The Yolov8n-D algorithm, a modification of the backbone part, improves mAP by 0.2% while reducing the parameters by 0.86 M, the computations by 2.2 G, and the model size by 1.4 MB, indicating a lighter and more efficient model structure without sacrificing accuracy. Similarly, the Yolov8n-S algorithm, focusing on the modified detection head, maintains the accuracy while reducing the parameters by 0.19 M, the computations by 1.1 G, and the model size by 0.3 MB through the Detect_SEAM algorithm. In comparison to the original Yolov8n algorithm, the final Yolov8n-FADS algorithm achieves a reduction of 1.1 M parameters, a 1.8 MB decrease in the model size, and a 4.9% improvement in the mAP. These results demonstrate that the improved model successfully accomplishes the objective of enhanced accuracy and reduced weight.

#### 4.3.3. Ablation Experiments

To further scrutinize the extent of the optimization of the detection performance of the original model through various improvement strategies, ablation experiments were conducted for comparative analysis under identical parameters. These experiments involved sequentially integrating modules, namely Yolov8n, Yolov8n-F for enhancing the loss function, Yolov8n-A-P2 for refining the head component, Yolov8n-D for enhancing C2_F in the backbone section, and Yolov8n-S for improving the detector head. The outcomes are summarized in Table 3, where the symbol ‘√’ denotes the incorporation of the respective method.

The experimental results presented in Table 3 demonstrate that the YOLOv8n algorithm outperforms the original Yolov8 algorithm following the integration of four enhancements into the training process. Specifically, the recall rate (R) increases by 10%, while the mean Average Precision (mAP) sees a boost of 6.5%. Moreover, there is a reduction of 1.1 M in the parameters and 1.8 MB in the memory space without any change in the GFLOPs. The efficacy of the proposed model in enhancing the performance of the original Yolov8n model is further validated through ablation experiments, underscoring the superiority and practical applicability of the proposed algorithm.

#### 4.3.4. Comparative Experiments

Various iterations of the prevalent object detection models, including Yolov3, Yolov5, Yolov6, Yolov7, and Yolov8, were chosen for comparison purposes. The hyperparameters and training parameters of these models were set to their default values, and they were trained on standardized datasets. The outcomes are presented in Table 4, demonstrating that the enhanced Yolov8n model enhances the detection accuracy by 6.5%. Additionally, considerations were made to optimize the model size, parameter count, and computational efficiency, while prioritizing both the detection accuracy and the rate.

Figure 7 depicts the training accuracy curves of both the Yolov8n and Yolov8n-FADS models. It is evident that during the initial training phase, the accuracy levels of the original model and the enhanced algorithm are comparable. However, beyond approximately 80 training iterations, the accuracy curve of the enhanced algorithm exhibits a steeper incline compared to that of the original algorithm. Consequently, the final accuracy attained by Yolov8n-FADS significantly surpasses that of the original algorithm.

### 4.4. Visualization and Analysis of Test Results

To enhance the evaluative approach of the Yolov8n-FADS algorithm’s performance, we present in Figure 8 the detection outcomes of both Yolov8n and the enhanced Yolov8n-FADS model on the test image dataset. These two cohorts serve as mutual controls, wherein the original model fails to detect helmet-wearing under the influence of the headlight, whereas the enhanced model succeeds. The subsequent cohort demonstrates the superior detection accuracy of the enhanced model. Our comparison reveals that the latter exhibits heightened contextual awareness and feature fusion capabilities, leading to more precise defect localization and identification, thereby enhancing the overall detection accuracy.

In order to further explore the improvement effect of Yolov8n-FADS, Yolov8n was selected. The Yolov8n-FADS method produces heat maps of the detected regions, where the lighter-colored parts indicate the lower confidence regions in the feature maps and the darker-colored parts are the higher regions. The results are presented in Figure 9. The Yolov8n-FADS algorithm focuses on feature regions that cover a wider area than those identified by Yolov8n, and it pays more attention to these regions. It is observed that the algorithm pays greater attention to the part of the head that is occluded by light, which addresses the issue of missed detection. It can be observed that the heat map of the enhanced algorithm encompasses a larger area and exhibits a more extensive distribution, which suggests the enhanced detection of the miner’s helmet.

## 5. Conclusions

We propose an algorithm for helmet recognition and detection based on an enhanced YOLOv8n to tackle the challenges arising from poor image quality and difficult feature extraction in underground environments. The Focal Loss enhances the model’s ability to detect uncommon target categories by adjusting the weights of positive and negative samples, prioritizing difficult-to-classify positive samples while reducing the weight of easily classifiable negative samples. The ASF-P2 structure effectively captures both global and local image feature information, thus enhancing the detection accuracy while minimizing the computational requirements and parameter count. Integration of the Dilated Reparam Block in UniRepLKNet and the improved RepNCSPELAN backbone in YOLOv9 enables the more efficient capture of spatially sparse features, enhancing the model’s ability to recognize complex patterns. Moreover, the SEAM module’s enhanced detection head, SEAMHead, improves the handling of occlusion issues.

The experimental results demonstrate that the modified algorithm offers several advantages, including reduced parameters, decreased computational expenses, heightened detection accuracy, and the ability to fulfill real-time demands. In comparison to the original model, the enhanced version exhibits a 29% reduction in the storage space, a decrease of 36.7% in the parameter count, and an improvement of 4.9% in the detection accuracy. Furthermore, compared to current models, our proposed approach achieves superior detection accuracy while simultaneously lowering the computational and storage capacity demands on the platform, thus facilitating easy deployment on resource-constrained devices.

## Figures and Tables

**Figure 1 sensors-24-03767-f001:**
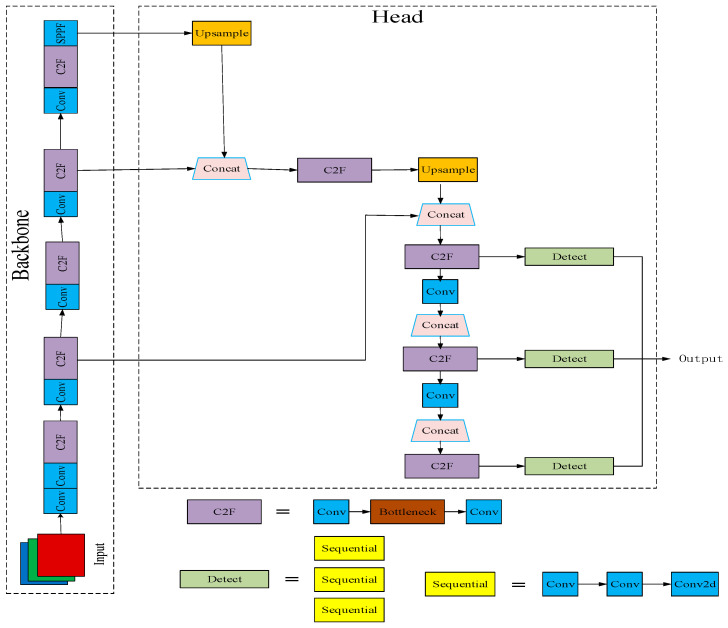
Structure of the Yolov8n model.

**Figure 2 sensors-24-03767-f002:**
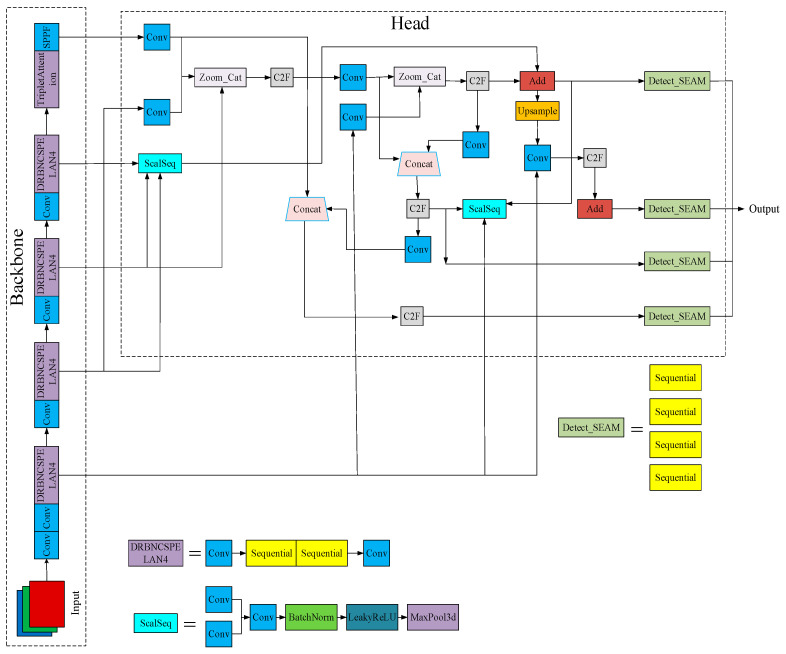
Structure of the Yolov8n-FADS model. In the Yolov8n configuration, the head processes feature maps using convolutional operations and dimensional mapping. In contrast, the Yolov8n-FADS configuration employs upsampling and concatenation to process feature maps, thereby affecting both the feature dimensions and the processing techniques.

**Figure 3 sensors-24-03767-f003:**
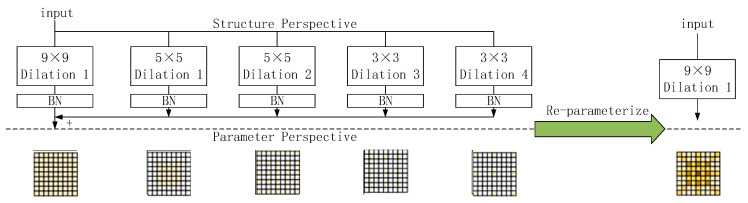
Structure of the Dilated Reparam Block.

**Figure 4 sensors-24-03767-f004:**
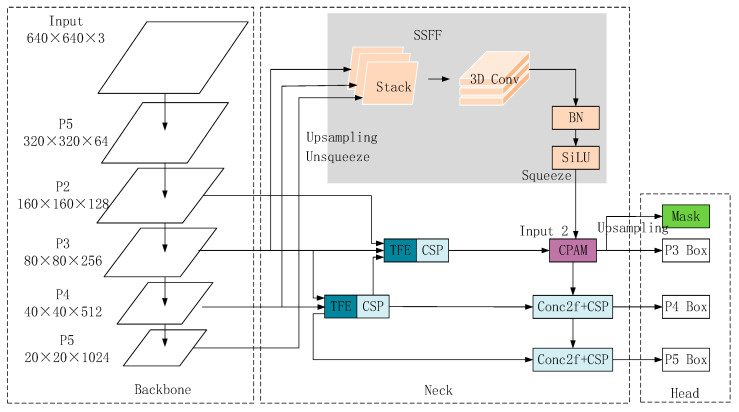
ASF-Yolo structure.

**Figure 5 sensors-24-03767-f005:**
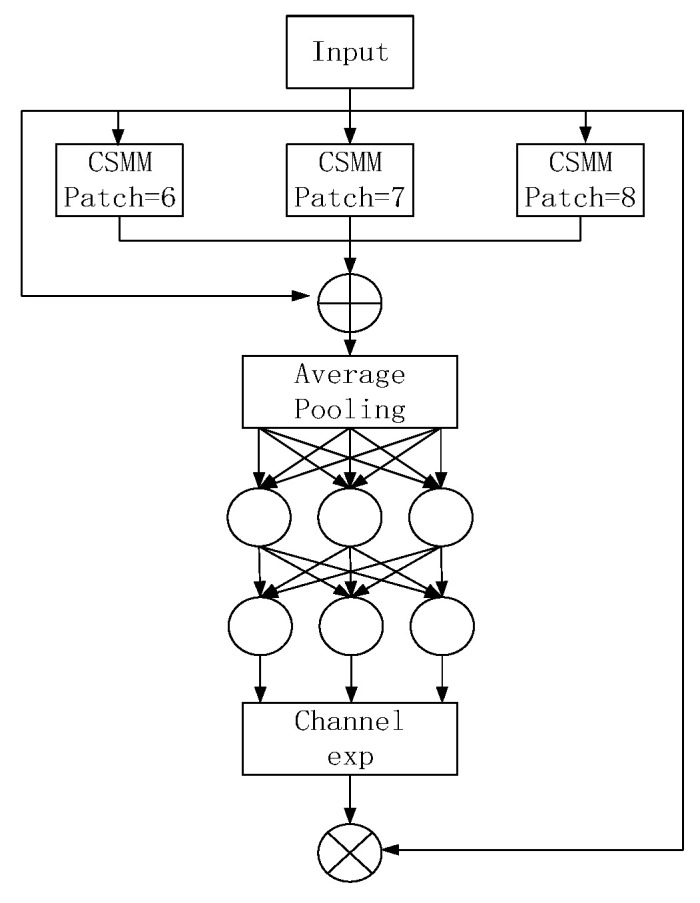
SEAM structure.

**Figure 6 sensors-24-03767-f006:**
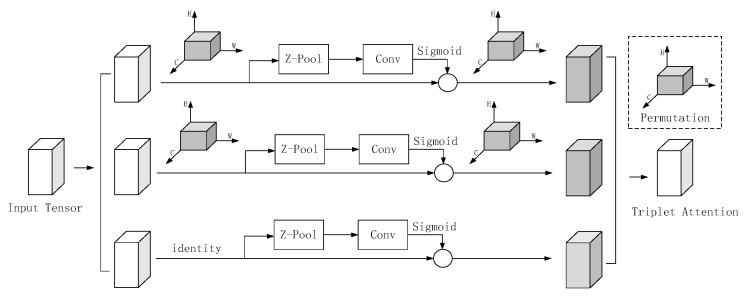
Structure of Triplet Attention.

**Figure 7 sensors-24-03767-f007:**
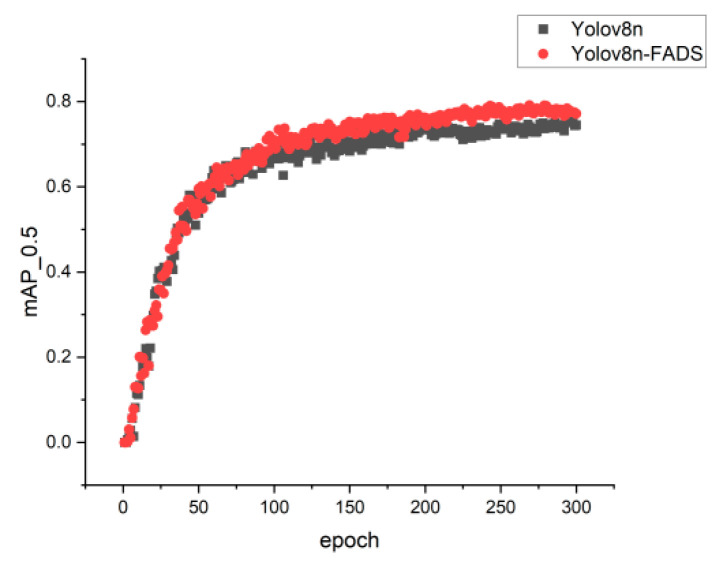
Comparison of the accuracy curves.

**Figure 8 sensors-24-03767-f008:**
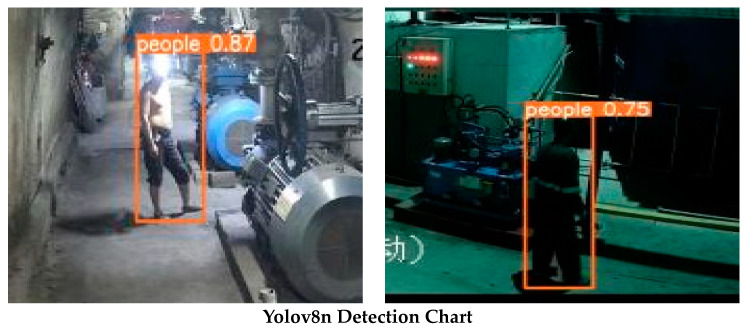
Comparison of the detection results.

**Figure 9 sensors-24-03767-f009:**
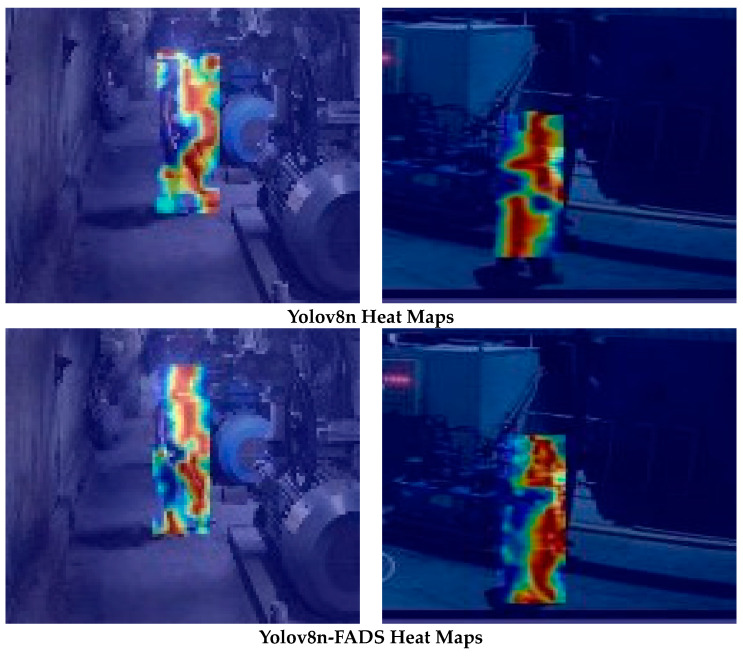
Comparison of the heat maps.

**Table 1 sensors-24-03767-t001:** Comparison of the objective indicators of the attention mechanisms.

Model Name	Params/M	FLOPs/G	Model/MB	mAP/%	Precision/ %	Recall/ %
Yolov8n	3.0	8.1	6.2	74.8	78	90.3
Yolov8n-MLCA	3.0	8.1	6.2	74.8	78	84.5
Yolov8n-SE	3.0	8.1	6.2	74.7	78	89.4
Yolov8n-SimAM	3.0	8.1	6.2	75.0	78	87.5
Yolov8n-Triplet	3.0	8.1	6.2	76.6	78	92.1

**Table 2 sensors-24-03767-t002:** Comparison of the experimental results.

Model Name	Params/M	FLOPs/G	Model/MB	mAP/%	Precision/ %	Recall/ %
Yolov8n	3.0	8.1	6.2	74.8	78	90.3
Yolov8n-F	3.0	8.1	6.2	76.5	80	89.9
Yolov8n-A-P2	2.49	12.0	5.4	79.7	86	86.6
Yolov8n-D	2.14	5.9	4.8	75.0	81	87.5
Yolov8n-S	2.81	7.0	5.9	74.8	79	92.4
Yolov8n-FADS	1.90	8.1	4.4	79.7	88	83.9

**Table 3 sensors-24-03767-t003:** Comparison of the ablation experiments.

Yolov8n	Yolov8n- F	Yolov8n-A-P2	Yolov8n-D	Yolov8n-S	Params/M	FLOPs/G	Model/MB	mAP/%
√					3.0	8.1	6.2	74.8
√			√		2.14	5.9	4.8	75.0
√		√	√		2.06	10.7	4.7	80.0
√		√	√	√	1.90	8.1	4.4	77.3
√	√	√	√	√	1.90	8.1	4.4	79.7

**Table 4 sensors-24-03767-t004:** Comparison of results of the different algorithms.

Model Name	Params/M	FLOPs/G	Model/MB	mAP/%
Yolov3-tiny	12.13	18.9	24.4	71.5
Yolov5n	2.5	7.2	5.3	70.8
Yolov6n	4.23	11.8	8.7	70.2
Yolov7-tiny	6.0	13.2	12.3	76.6
Yolov8n	3.0	8.1	6.2	74.8
Yolov8n-FADS	1.90	8.1	4.4	79.7

## Data Availability

Te datasets generated during and/or analysed during the current study are available from the corresponding author on reasonable request.

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
