# Peer review of "Yolov8n-FADS: A Study for Enhancing Miners’ Helmet Detection Accuracy in Complex Underground Environments"

_sensors, 2024, doi:10.3390/s24123767_

Round 1

Reviewer 1 Report

Comments and Suggestions for Authors

Please address these questions both by responding and by modifying the paper.

In the abstract there is no mention at all of the name of the method, which seems to be "Yolov8n-FADS".  Is this true? 

ASF should be explained in the abstract.

The section numbers start at 0 which seems odd.  I am going to reference sections based on starting at 0 in the review here.

The introduction is not well organized; it explains non-deep learning methods for helmet detection, describes general 2-stage detection with no references to helmet detection, then briefly describes yolo and follows with descriptions of yolo modifications for helmet detection.  There is then more extensive discussion of YoloV8 in section 1, except section 1 also seems to reference yolov9 and modifications made for yolov8n-FADs, and then Yolov8n-FADs is described in section 2.

These sections need to be re-organized up to section 2.

There needs to be more motivation as to why the modifications were made and what they mean with respect to the helmet detection problem; it may be that the modifications are not specifically for helmet detection, but it is not clear.

The figure captions need more descriptions.  "structure of the yolov8n model" does not really tell me anything; describe the sections (briefly) in the captions so I can learn more from the figure.

Most parts in section 2 seems to try and indicate what the improvement is, but it does not explain why it is helpful for helmet detection nor HOW the improvement works.

The loss function section is better, but what is 'edge regression'?

Line 153, the focaler should be superscript throughout.

Line 166, it says "To compensate for this loss' which is confusing since loss means something specific for deep network training.  Maybe "this issue" would be better?

A reference for Labelimg should be provided.

It is hard to compare the tabulated results; table 1 and 2 have the same caption.  Table 1 seems to be comparing the differnt attention mechanisms, while table 2 seems to be comparing something else but it is not clear.  An explanation of the steps taken and planned comparisons would be very helpful.  Graphs or plots would help as well, I think. 

Figures 7 & 8 are hard to see; perhaps zooming in on the detections would help.

The first sentence of the conclusion is not well supported.  "Solves the problem" is a bold absolute statement esp when applied to 'object detection' broadly.  

What are spatially sparse distributed features?

Comments on the Quality of English Language

Some English problems exist, although overall most of the text is understandable.  I recommend an editing with an English-language subject matter expert.

Author Response

Point-by-point response to Comments and Suggestions for Authors

Thank you for your review of our paper. We have answered each of your points below.

Comments 1: [In the abstract there is no mention at all of the name of the method, which seems to be "Yolov8n-FADS". Is this true?]

Response 1: [In the abstract it is written that the proposed algorithm is Yolov8n-FADS. modifications have been made, rewritten and marked in yellow. P.1 yellow marker text]

Comments 2: [ASF should be explained in the abstract.]

Response 2: [ASF is an Attention Scale Sequence Fusion framework that combines spatial and scale features for accurate and fast segmentation of cell instances. An explanation is given in the abstract.]

Comments 3: [The section numbers start at 0 which seems odd. I am going to reference sections based on starting at 0 in the review here.]

Response 3: [Sorry, I've changed the chapter numbers. The suggestions you made based on the chapters have also been changed.]

Comments 4: [The introduction is not well organized; it explains non-deep learning methods for helmet detection, describes general 2-stage detection with no references to helmet detection, then briefly describes yolo and follows with descriptions of yolo modifications for helmet detection. There is then more extensive discussion of YoloV8 in section 1, except section 1 also seems to reference yolov9 and modifications made for yolov8n-FADs, and then Yolov8n-FADs is described in section 2.

These sections need to be re-organized up to section 2.]

Response 4: [Dear reviewers, thank you for your careful review and constructivesuggestions regarding our manuscript. We have revised the manuscript inaccordance with the comments and marked all the amends on our revisedmanuscript.

In the introduction section the significance of helmet detection is added and then a diagram of the network structure of Yolov8n is added so that it can be compared with that of Yolov8n-FADS, which makes it more straightforward and clear to understand how the improvements were made. This section of the Yolov8n-FADS detection model explains several of the improvements that were used, and a description of which are described and illustrated, which can be seen to be useful for helmet detection through the table of experimental results.

Is this modification ok, thanks again for the suggestion, if there are any other issues I will make more detailed modifications. ]

Comments 5: [There needs to be more motivation as to why the modifications were made and what they mean with respect to the helmet detection problem; it may be that the modifications are not specifically for helmet detection, but it is not clear.]

Response 5: [I agree with this comment. In accordance with your recommendation, I have incorporated a discussion of the significant role of helmet testing and related matters in the introductory section.]

Comments 6: [The figure captions need more descriptions. "structure of the yolov8n model" does not really tell me anything; describe the sections (briefly) in the captions so I can learn more from the figure.]

Response 6: [I've added a diagram of the network structure of Yolov8n to the introduction of Yolov8, and this can be compared with the network structure of Yolov8n-FADS to get a clearer picture of where the changes have been made, and which parts have been improved.]

Comments 7: [Most parts in section 2 seems to try and indicate what the improvement is, but it does not explain why it is helpful for helmet detection nor HOW the improvement works.]

Response 7: [I have moved the network structure diagram and the introduction of Yolov8n-FADS to the Yolov8n-FADS detection model section, and the comparison of the 2 network structure diagrams is the part of the improvement that can be seen. This section of the Yolov8n-FADS detection model explains several of the improvements that were used, and a description of which are described and illustrated, which can be seen to be useful for helmet detection through the table of experimental results.]

Comments 8: [The loss function section is better, but what is 'edge regression'?]

Response 8: [Please accept my apologies for the error in the original text, which should have read "boundary box regression." I have made the necessary correction. P.4 2.3 Part yellow marker text]

Comments 9: [Line 153, the focaler should be superscript throughout.]

Response 9: [I apologize for the oversight. The necessary changes have been implemented.]

Comments 10: [Line 166, it says "To compensate for this loss' which is confusing since loss means something specific for deep network training. Maybe "this issue" would be better?]

Response 10: [ While depth-separated convolution understands the importance of different channels and reduces the number of parameters, it ignores the informative relationships between channels. The loss refers to this informational relationship that it ignores. Loss does have specific implications for deep network training. Maybe my expression is wrong, is it ok if I replace it with the word defective.]

Comments 11: [A reference for Labelimg should be provided.]

Response 11: [Labeling is an image annotation tool. Despite an exhaustive search, I was unable to locate its original documentation. Is it possible to not make a citation.]

Comments 12: [It is hard to compare the tabulated results; table 1 and 2 have the same caption. Table 1 seems to be comparing the differnt attention mechanisms, while table 2 seems to be comparing something else but it is not clear. An explanation of the steps taken and planned comparisons would be very helpful. Graphs or plots would help as well, I think.]

Response 12: [Yes, Table 1 compares the results of the different attention mechanisms and Table 2 lists the results of the different enhancement methods used. I have explained the comparisons made in Table 2, which are highlighted in yellow in the text. page 8]

Comments 13: [Figures 7 & 8 are hard to see; perhaps zooming in on the detections would help.]

Response 13: [Figures 7 and 8 have been enlarged and resized for a more detailed and clearer view.]

Comments 14: [The first sentence of the conclusion is not well supported. "Solves the problem" is a bold absolute statement esp when applied to 'object detection' broadly.]

Response 14: [I agree with this comment. A revised description of the initial sentence in the conclusion of the paper has been provided.]

Comments 15: [What are spatially sparse distributed features?]

Response 15: [Spatially sparse distributed features refer to patterns or characteristics that are not densely concentrated in a specific area or region, but rather are spread out or distributed sparsely across a spatial domain. In the context of data analysis and signal processing, spatially sparse distributed features often imply that the relevant information or attributes are not densely clustered in a particular location, but instead exhibit a more scattered or sparse distribution across the spatial domain.]

Comments on the Quality of English Language
We tried our best to improve the manuscript and made some changes to the manuscript. These changes will not influence the content and framework of the paper. And here we did not list the changes. We appreciate for Editors/Reviewers' warm work earnestly and hope that the correction will meet with approval.

Reviewer 2 Report

Comments and Suggestions for Authors

Dear authors,

Thank you for an interesting article. The text describes in detail the structure and dependencies between the individual modules of the YOLO algorithm. The article contains extensive literature review and a detailed introduction to the discussed topic. The obtained results were compared with other known versions of the Yolo algorithm (as can be seen in Table 4), and as can be seen, these results are very satisfactory.

The results of your work may be particularly important when the algorithm is applied to simple computers that do not have high computing power. This is especially important in industrial applications, where it is important to maximize effects while minimizing costs.

In order to expand the group of recipients of your research results and to make the obtained results also available to engineers who can influence the possible application of the solution in practice, I propose:

- increase the readability of drawings no. 7 and 8 - perhaps the solution will be to use two rather than three views (the first two are similar anyway), which will allow to increase the size of the drawings,

- heatmaps visualization has been introduced in Figure No. 8, this aspect is discussed in the text in lines 296-298, however, the text does not provide direct results as to what the purpose of this type of presentation is, and how to interpret the color maps in Figure No. 8 - for To increase the clarity of the text, it is necessary to enter an additional description in the text for this figure.

Author Response

Point-by-point response to Comments and Suggestions for Authors

Thank you for your review of our paper. We have answered each of your points below.

Comments 1:
[increase the readability of drawings no. 7 and 8 - perhaps the solution will be to use two rather than three views (the first two are similar anyway), which will allow to increase the size of the drawings.]

Response 1: [I agree with this comment. The second image in Figure 7 and Figure 8 was deleted, and the image size was increased.]

Comments 2: [heatmaps visualization has been introduced in Figure No. 8, this aspect is discussed in the text in lines 296-298, however, the text does not provide direct results as to what the purpose of this type of presentation is, and how to interpret the color maps in Figure No. 8 - for To increase the clarity of the text, it is necessary to enter an additional description in the text for this figure.]

Response 2: [I agree with this comment. A text note has been added to page 11 to elucidate the rationale behind the incorporation of the heat map. The enhanced algorithm's detection area is comparatively expansive.  P.11 yellow marker text]

Round 2

Reviewer 1 Report

Comments and Suggestions for Authors

The manuscript is improved but there are still some issues.  The Chinese characters for 'References' should be replaced by English.  LIne 97, give a reference for yolov8 / ultralytics (URL is probably best).  A brief sentence pointing out the head differences between yolov8n and yolov8-fads would be nice for figure 2 caption.  I do not feel like point 5 is adequately addressed; are the modifications specifically for helmet detection (I don't think they are, rather, the researchers are motivated specifically for helmet detection).  So lines 82-91 would be a good candidate to state this.  The captions still need to be improved - some readers (like me) tend to first skim through, look at the paper structure, and having information in the caption that tells me a little about the significance of the figure is helpful.   For labelimg, is this the tool?  https://pypi.org/project/labelImg/  

Author Response

Point-by-point response to Comments and Suggestions for Authors

Thank you for your review of our paper. We have answered each of your points

below.

Comments 1: [The Chinese characters for 'References' should be replaced by English.]

Response 1: [I extend my apologies for this error and have implemented the necessary corrections. The relevant text has been highlighted in grey.]

Comments 2: [LIne 97, give a reference for yolov8 / ultralytics (URL is probably best).]

Response 2: [Added references in URL format. Glenn Jocher and Ayush Chaurasia and Jing Qiu. Ultralytics YOLOv8. 2023. https://github.com/ultralytics/ultralytics]

Comments 3: [A brief sentence pointing out the head differences between yolov8n and yolov8-fads would be nice for figure 2 caption.]

Response 3: [The title of Fig 2 has been revised to concisely delineate the disparity between the two.]

Comments 4: [I do not feel like point 5 is adequately addressed; are the modifications specifically for helmet detection (I don't think they are, rather, the researchers are motivated specifically for helmet detection). So lines 82-91 would be a good candidate to state this.]

Response 4: [I'm sorry, it really didn't address the issue you raised last time. Ensuring that the motivation and scope of the changes are clearly communicated is essential and has been stated at the end of lines 82-91. It can be widely used for other target detection tasks.]

Comments 5: [The captions still need to be improved - some readers (like me) tend to first skim through, look at the paper structure, and having information in the caption that tells me a little about the significance of the figure is helpful.]

Response 5: [Changes have been made to the title to reveal some important information.]

Comments 6: [For labelimg, is this the tool?  https://pypi.org/project/labelImg/  ]

Response 6: [Yes, LabelImg is a popular image annotation tool that helps users to annotate images and generate datasets for training target detection models. More information about LabelImg, including installation guides, usage instructions, and other related resources, can be obtained from this link. The dataset in the article is the public dataset CUMT-HelmeT from the China University of Mining and Technology, which I annotated using LabelImg.]

 Thank you for your valuable feedback.
